# *In silico* analysis identifies a putative cell-of-origin for *BRAF* fusion-positive cerebellar pilocytic astrocytoma

Subhi Talal Younes[1,2]*, Betty Herrington[3]

**1** MD/PhD Program, University of Mississippi Medical Center, Jackson, Mississippi, United States of America, **2** Department of Physiology and Biophysics, University of Mississippi Medical Center, Jackson, Mississippi, United States of America, **3** Division of Pediatric Hematology/Oncology, Department of Pediatrics, University of Mississippi Medical Center, Jackson, Mississippi, United States of America

* styounes@icloud.com

## Abstract

Childhood cancers are increasingly recognized as disorders of cellular development. This study sought to identify the cellular and developmental origins of cerebellar pilocytic astrocytoma, the most common brain tumor of childhood. Using publicly available gene expression data from pilocytic astrocytoma tumors and controlling for driver mutation, a set of developmental-related genes which were overexpressed in cerebellar pilocytic astrocytoma was identified. These genes were then mapped onto several developmental atlases in order to identify normal cells with similar gene expression patterns and the developmental trajectory of those cells was interrogated. Eight known neuro-developmental genes were identified as being expressed in cerebellar pilocytic astrocytoma. Mapping those genes or their orthologs onto mouse neuro-developmental atlases identified overlap in their expression within the ventricular zone of the cerebellar anlage. Further analysis with a single cell RNA-sequencing atlas of the developing mouse cerebellum defined this overlap as occurring in ventricular zone progenitor cells at the division point between GABA-ergic neuronal and glial lineages, a developmental trajectory which closely mirrors that previously described to occur within pilocytic astrocytoma cells. Furthermore, ventricular zone progenitor cells and their progeny exhibited evidence of MAPK pathway activation, the paradigmatic oncogenic cascade known to be active in cerebellar pilocytic astrocytoma. Gene expression from developing human brain atlases recapitulated the same anatomic localizations and developmental trajectories as those found in mice. Taken together, these data suggest this population of ventricular zone progenitor cells as the cell-of-origin for *BRAF* fusion-positive cerebellar pilocytic astrocytoma.

## Introduction

A developmental origin of childhood cancer is well recognized [1]. For example, the neoplastic cells which give rise to pediatric leukemia are often present at birth, years before manifestation

**Data Availability Statement:** All relevant data are within the paper and its supporting information files.

**Funding:** The authors received funding for open access fees from the following: NIH grants

P20GM104357 (University of Mississippi Medical Center COBRE) and U54GM115428 (Mississippi Center for Clinical and Translational Research).

**Competing interests:** The author declares that no competing interests exist.

of disease [2–6]. Moreover, the mutations occurring within childhood cancers often inhibit cellular differentiation and treating the neoplastic cells with agents which induce differentiation has proven to be a highly effective therapeutic approach [7, 8]. Thus, understanding the developmental processes which have gone awry during tumorigenesis is crucial to understanding the biology of pediatric tumors and may inform therapeutic approaches.

Tumors of the central nervous system (CNS) are the most common solid malignancy of childhood and are the leading cause of cancer-related deaths in children and adolescents [9, 10]. Moreover, many of those children who are cured must confront and manage treatment-related morbidity due to toxicity associated with contemporary radiation and chemotherapy treatment regimens [11–13].

Spatiotemporal restriction of driver mutations in pediatric CNS tumors suggests that these mutations are only oncogenic within certain cellular contexts [14, 15]. As such, pediatric CNS cancer is widely recognized to be a disorder of neural development, whereby oncogenic mutations hijack normal developmental pathways within the cell-of-origin to drive tumor initiation, growth, and progression [16]. For example, medulloblastoma is now understood to represent a heterogenous disease with distinct cellular and developmental origins [15, 17, 18]. Treatment approaches are now stratified based upon these distinctions [19], underscoring the importance of understanding the developmental biology and cell-of-origin of pediatric CNS tumors.

Cerebellar pilocytic astrocytoma is the most common CNS tumor in children, with over 500 being diagnosed in the United States each year [10]. Though complete resection is curative, surgery is often associated with significant morbidity, and not all tumors are amenable to surgery, necessitating the use of adjuvant radiation or chemotherapy with resultant increased risk of aforementioned treatment-related morbidity [12, 13]. Further study of the underlying biology of cerebellar pilocytic astrocytoma is needed in order to inform improved therapeutic approaches.

By far, the most common driver mutation in cerebellar pilocytic astrocytoma is fusion of the *BRAF* locus with a variety of fusion partners, the most frequent being *KIAA1549* [20–23]. The common effect of these fusions is constitutive activation of the MAPK pathway with resultant driving of cellular growth and division [24]. Whereas this molecular signaling has been well studied, the cellular and developmental origins of cerebellar pilocytic astrocytoma remain shrouded in mystery. The objective of this study was to elucidate and define the cellular and developmental origins of pediatric cerebellar pilocytic astrocytoma. The central hypothesis of this study is that cerebellar pilocytic astrocytoma arises from a distinct cell-of-origin within the developing brain and that the developmental gene expression program thereof is retained within the tumor cells.

## Materials and methods

### Microarray gene expression analysis

Gene expression data was obtained from the Gene Expression Omnibus (GEO) in the form of a GEO dataset using the accession number GSE44971 [25] and the R package GEOquery [26]. The rest of the data analysis was performed using a variety of R packages [27–34]. Samples without a documented BRAF fusion were removed from the dataset. Two infratentorial tumors with a BRAF fusion were removed since their location was listed as "Brainstem" and "fourth ventricle" rather than "cerebellum." Gene expression contrast was run using a linear model as implemented in the R package limma [35] for infratentorial vs. supratentorial tumors. The resulting p-value was adjusted for multiple testing via the Benjamini and Hochberg method. A P value of $< 0.001$ was considered to be statistically significant.

For the heatmap, a distance matrix was constructed using the non-centered Pearson method $[(1 - sum(x\_i \, y\_i) / sqrt [sum(x\_i\textasciicircum 2) \, sum(y\_i\textasciicircum 2)]$ as implemented in the R package amap [36]. Unsupervised hierarchical clustering using the complete linkage method was then performed on the distance matrix. Gene expression data was viewed globally using the Barnes-Hut implementation of t-weighted stochastic neighbor embedding (t-SNE) wrapped in the package Rtsne [34] with the following non-default parameters: perplexity = 12, theta = 0, max_iter = 2500.

## Gene ontology analysis

Gene ontology analysis was conducted as described in the Bioconductor workflow "maEnd-toEnd" [37]. Briefly, genes with similar absolute expression intensities to genes which were differentially expressed between infratentorial and supratentorial tumors were selected as a background set. Thus, the gene universe consisted of differentially expressed genes and a background set of genes with similar expression intensities. The package topGO [31] was then applied to this dataset using the "Biological Processes" ontology mode and Fisher exact testing for statistical significance. Gene annotation data was imported from the Affymetrix Human Genome U133 Plus 2.0 Array annotation data [28]. The top five gene ontology terms were reported.

## Selection of genes for analysis

Similar to the method described by Gibson, et al. [15], genes which were overexpressed in *BRAF* fusion-positive infratentorial tumors were selected as candidate marker genes for the cell-of-origin. Then, the Allen Developing Mouse Brain Atlas (Allen Institute; 2008) and the Cell Seek single cell RNA sequencing atlas [38] were interrogated for these genes or their mouse orthologs. Several genes, though present in the one or both atlases, exhibited no expression (Ppargc1A, Ntrk3, Camk4, Gdf11) or nonspecific expression (i.e. all cell types) across the entire brain and/or developing cerebellum (Cntn1, Cntn3, Tsc22D1, Gria4). As a result, these genes were excluded from subsequent analysis. The remaining genes (Ascl1, Irx2, Irx5, Klf15, Meis1, Msx2, Pax3, and Pbx3) were validated in an independent pilocytic astrocytoma gene expression dataset and used for downstream analysis. These genes are hereafter referred to as pilocytic astrocytoma-developmentally related (PA-DR) genes.

## Gene expression mapping using Allen Developing Mouse Brain Atlas

Because data on gene expression in the developing human cerebellum is sparse while mouse gene expression atlases are more robust, analysis was first performed using data from mice. Brain Explorer 2 is a desktop software application which can visualize the Allen Developing Mouse Brain Atlas gene expression data in three dimensions. The software also allows highlighting of particular anatomic structures. The methodology used to generate this and map this data is provided by the Allen Institute for Brain Science at http://developingmouse.brain-map.org/content/explorer. Data for all eight PA-DR genes at embryonic day 13.5 and 15.5 were interrogated as shown in S2 Fig. Images were taken in the sagittal and coronal anatomic planes. Rhombomere 1 was highlighted in each captured image as was the cerebellar anlage. For the summary data shown in Fig 2A and 2B, high resolution ISH images from the Developing Mouse Brain Atlas were accessed for each PA-DR gene. Using the associated reference atlas, each gene was manually called to be expressed or not in the ventricular zone and the external granule cell layer. Individual microscopic images used for this analysis are available in S1 Data while the results are available in S2 Table.

## Cell seek analysis

The Cell Seek database was accessed at https://cellseek.stjude.org/cerebellum. Data for each of the PA-DR genes was accessed and an image of the resulting expression data was taken. The overlap image shown in Fig 3A was generated using the GNU Image Manipulation Program (https://www.gimp.org) in the following manner. The images for the top four or all PA-DR genes was loaded into GIMP as a layer. Each layer was made transparent to 25% and merged with the layer mode set to "Merge." Selection of cells for subsequent lineage analysis was guided to incorporate each of the major cell types seen within the cluster enriched for PA-DR genes (see Fig 3A). Monocle, BEAM, and transcription factor correlation were performed using the built in functions of Cell Seek [38]. For details of the implementation therein, see reference [38]. The q-value threshold for BEAM was set at 0.001.

## Expression in developing human brain

The BrainSpan atlas of the developing human brain [39] (http://www.brainspan.org) was used to interrogate the expression of PA-DR genes therein. The resulting heatmap was sorted by structure. For organoid expression data, the scApeX portal from the accompanying publication [40] was accessed (https://bioinf.eva.mpg.de/shiny/sample-apps/scApeX/). The expression of each PA-DR gene was interrogated and reported as it resulted in the portal (i.e. no further analysis or alteration was done).

# Results

## Gene expression contrast of BRAF fusion-positive infratentorial pilocytic astrocytoma identifies neural development related genes

In order to identify the developmental origins of cerebellar pilocytic astrocytoma, a publicly available gene expression dataset was used which contains information from 49 pilocytic astrocytomas and 9 normal cerebellar tissues [25]. Two principles drove subsequent analysis. First, presumably the most common driver mutation in pilocytic astrocytoma (i.e. *BRAF* fusion) produces a similar set of gene expression changes regardless of tumor location/cell of origin. Second, contrasting gene expression between tumors of differing location/cell of origin but with identical driver mutation should mask those changes induced by the *BRAF* fusion, leaving only those genes expressed in the presumptive cell of origin (i.e. those genes which are unique to that tumor location/cell of origin). Therefore, the tumor dataset was filtered to include only those tumors with a documented *BRAF* fusion, regardless of fusion partner. This left a set of 37 tumors, 5 supratentorial and 32 infratentorial. All subsequent analyses were performed on this set.

Consistent with the overarching hypothesis, dimensionality reduction failed to distinguish between supra- and infra-tentorial tumors (Fig 1A and 1B), suggesting that *BRAF* fusion drives similar gene expression changes in both groups of tumors [41, 42]. Despite this similarity, gene expression contrast between infra- and supra-tentorial tumors identified multiple differentially expressed genes. This data suggests that these differentially expressed genes represent those genes expressed in the cell of origin for each respective tumor location. Indeed, consistent with prior reports [25] (albeit ones which included non-*BRAF* translocated tumors), several neuro-developmental related genes were found to be overexpressed in infratentorial tumors (Fig 1C), and top gene ontology terms for differentially expressed genes were largely related to cellular/neural development (Fig 1D). This result demonstrates the potential utility of contrasting tumors with identical driver mutations but differing anatomic locations in order to reveal the underlying developmental program operative in the cell of origin.

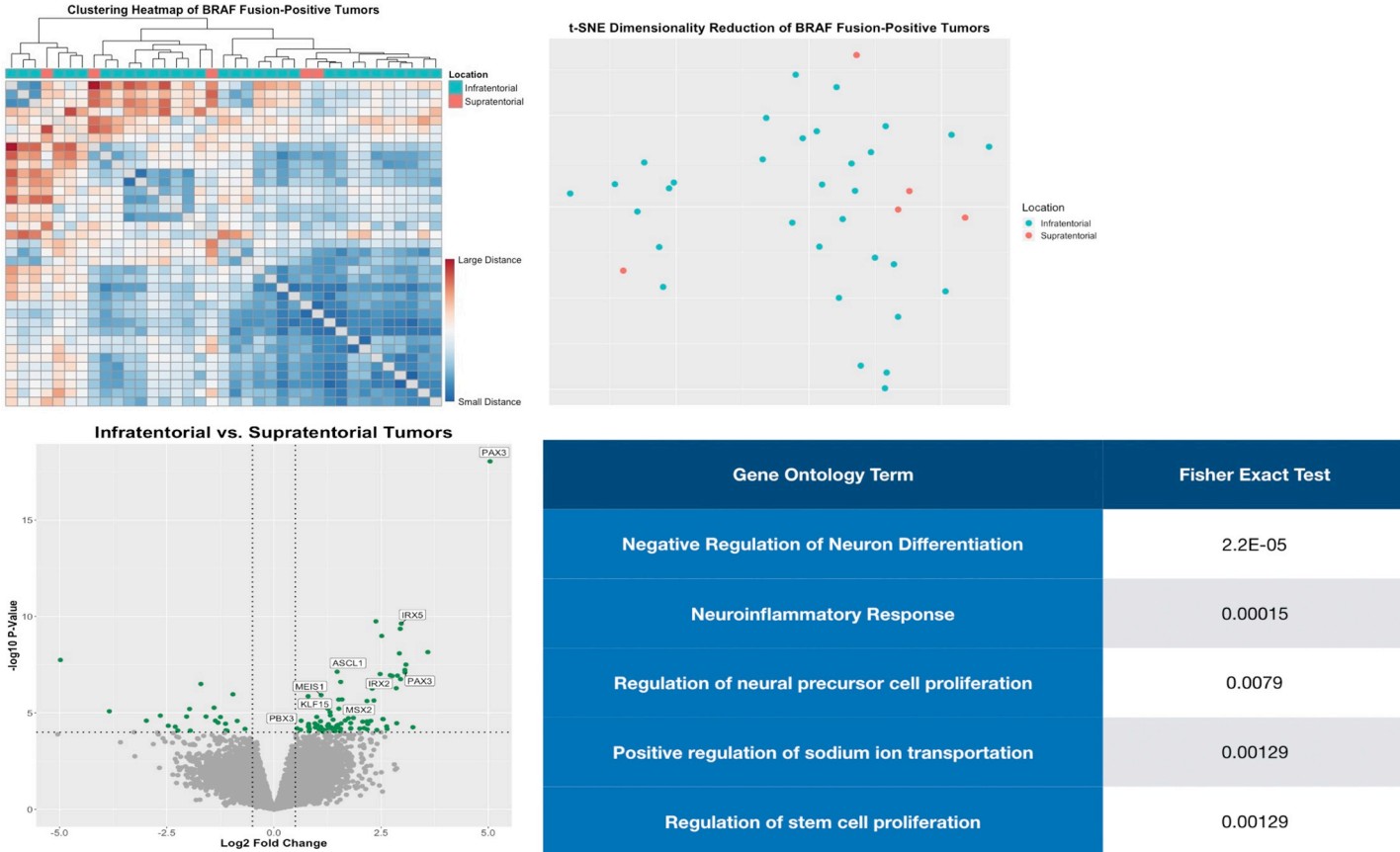

**Fig 1. Gene expression of *BRAF* fusion-positive pilocytic astrocytoma identifies several neural developmental-related genes in infratentorial tumors.** (A) Heatmap of global gene expression data from all *BRAF* fusion-positive tumors in the cohort. (B) T-weighted stochastic neighbor embedding (t-SNE) plot of gene expression from *BRAF* fusion-positive tumors. (C) Volcano plot of gene expression data from infratentorial vs. supratentorial *BRAF* fusion-positive pilocytic astrocytomas. Known developmental genes which were found to be differentially expressed are labeled (see text). (D) Gene ontology terms for differentially expressed genes in infratentorial vs. supratentorial *BRAF* fusion-positive pilocytic astrocytomas. P-value represents that derived from a Fisher's exact test.

Of note, very few genes were differentially expressed, with only 97 being significantly upregulated and 26 significantly downregulated (S1 Table). This is in keeping with the relatively quiescent genomic perturbances which characterize pilocytic astrocytoma [25, 43, 44].

## Neural development related genes overexpressed in infratentorial pilocytic astrocytoma are co-expressed in the ventricular zone of the developing cerebellum

Similar to the methodology described by Gibson, et al. [15], genes overexpressed in infratentorial tumors or their mouse orthologs were cross-referenced with the Allen Developing Mouse Brain Atlas (www.brain-map.org, Allen Institute) in order to identify the anatomic origins of cerebellar pilocytic astrocytoma. Using *in situ* hybridization, this resource catalogs expression of over 2000 genes within the developing mouse brain across 7 developmental stages. Of 97 upregulated genes in infratentorial tumors (i.e. those genes expressed in infratentorial but not supratentorial pilocytic astrocytoma), 16 were found to be present in the atlas. Of these, four genes exhibited no expression (Ppargc1A, Ntrk3, Camk4, Gdf11) across the developing brain and four (Cntn1, Cntn3, Tsc22D1, Gria4) exhibited diffuse expression across the entire brain and/or developing cerebellum. As a result, these genes were excluded from

subsequent analysis. This left eight genes to be analyzed (Ascl1, Irx2, Irx5, Klf15, Meis1, Msx2, Pax3, and Pbx3). In order to validate these genes, an independent dataset of gene expression from pilocytic astrocytoma tumors was interrogated [45]. All eight of these genes were reported to be overexpressed in infratentorial as compared to supratentorial tumors in this independent dataset, validating these findings. Hereafter, this group of genes is referred to as pilocytic astrocytoma development-related (PA-DR) genes.

First, the temporal expression pattern of each of these eight genes was interrogated. The temporal patterns of PA-DR gene expression differed depending on the particular gene in question (S1 Fig). However, embryonic days 13.5 and 15.5 emerged as being the time point in development with the most significant overlap in expression amongst all PA-DR genes. Therefore, these time points were selected for further analysis.

To further annotate anatomic localization, this data was visualized using Brain Explorer 2, an application which plots the *in situ* hybridization data of the Allen Developing Mouse Brain Atlas in three dimensions. Six out of eight PA-DR genes exhibited expression within rhombomere 1, the embryological structure which gives rise to the cerebellum, on both embryonic day 13.5 and 15.5 (S2A to S2B Fig). Furthermore, many PA-DR genes exhibited overlap in their localization thereof, suggesting cellular and/or regional overlap in expression. This suggests that cerebellar pilocytic astrocytoma arises from a cell within the cerebellar anlage, unlike some cerebellar tumors which have origins outside the cerebellum proper (e.g. Wnt-subtype medulloblastoma) [15].

The many different cell types of the adult cerebellum can be conceived of as arising from two distinct developmental pathways. Glutamatergic neurons arise from the external granule cell layer, while GABA-ergic neurons as well as cerebellar glial cells arise from the ventricular zone (see excellent review by Martinez et al., 2013) [46]. Note that the neurons of the deep cerebellar nuclei also arise from the ventricular zone with a stereotypic migration pattern from the ventricular zone to a structure known as the nuclear transitory zone. Thus, further analysis was conducted to identify which of these embryologic structures are responsible for the gene expression overlap described above.

As shown in Fig 2A and 2B, the ventricular zone of the cerebellar anlage exhibited significantly more overlap in expression of PA-DR genes than the external granule cell layer (see S2 Table and S1 Data), suggesting a ventricular zone origin for *BRAF* fusion-positive cerebellar pilocytic astrocytoma.

## PA-DR genes are co-expressed in early ventricular zone progenitors and straddle the differentiation point between GABA-ergic neuron and glial progenitor cells

Since multiple cell types are present within the ventricular zone, further definition of the individual cells which display co-expression of PA-DR genes was needed. To accomplish this, Cell Seek, a single cell RNA sequencing dataset of the developing mouse cerebellum [38] was used. As shown in Fig 3A, a small population of cells exhibited overlap in expression of PA-DR genes. In particular, Pax3 and Ascl1 exhibited highly specific expression in the Cell Seek database (S3A Fig). These cells are isolated from the embryologic day 13, 14, and 15 (S3B Fig) and are classified by the Cell Seek software as "progenitor," "GABA progenitor," "glia," and "astrocytes," based upon their expression of known cellular markers (S3C Fig) [38].

In order to further define the cellular and developmental trajectory of this cell population, 2,262 cells compromising the aforementioned four major lineages were selected to perform further analysis upon (S4A Fig). Lineage analysis with Monocle identified a cluster of early precursor cells with three different developmental trajectories (Fig 3B). The first lineage decision

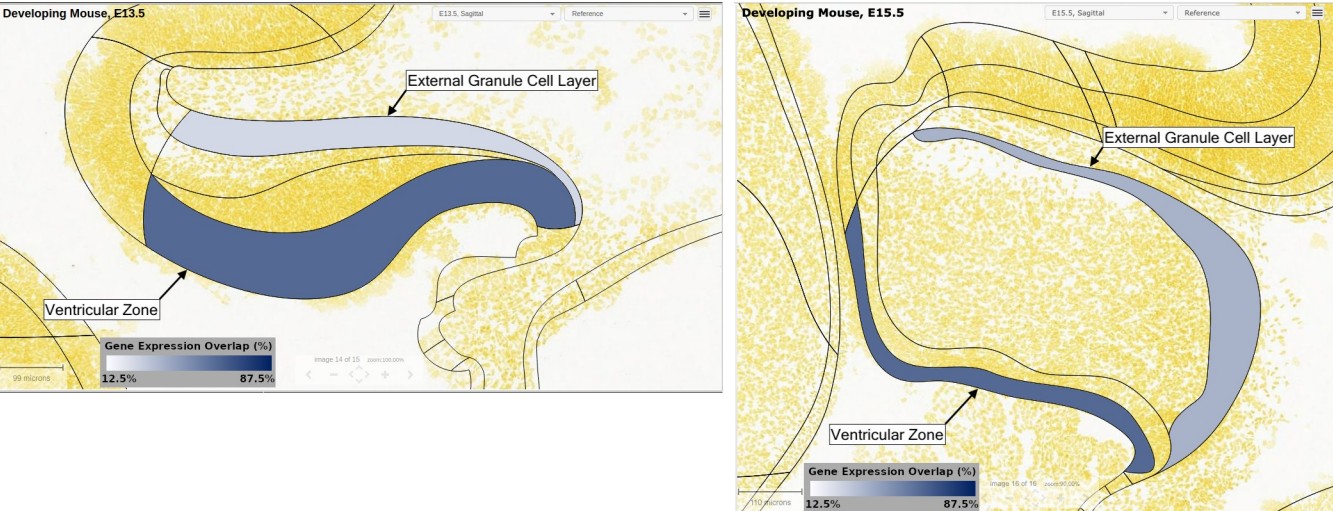

**Fig 2. Neurodevelopmental-related genes overexpressed in infratentorial pilocytic astrocytoma are co-expressed in the ventricular zone of the developing cerebellum.** (A-B) Summary data indicating percentage of PA-DR genes expressed within the indicated anatomic zone on gestation day 13.5 (A) and 15.5 (B) demonstrating that PA-DR genes are enriched within the ventricular zone. Images represent sagittal sections of the developing mouse cerebellum from the Allen Brain Reference Atlas. Cell nuclei are stained yellow. The regions outlined by black lines represent different developmental structures. The external granule cell layer and ventricular zone layer are labeled and colored based on the percentage of PA-DR genes expressed in that region. See also S2 Table and S1 Data.

was between cells with markers of the early GABA-ergic lineage (i.e. Ptf1a) and a lineage characterized by stem cell/neuronal progenitor markers (e.g. Pax6 and Otx2) (Fig 3C). This latter cell lineage was further divided into astrocytes and a cryptic group of cells which express some markers of glutamatergic cell lineage (e.g. Pax6, Id1, Ybx1), yet lack expression of canonical glutamatergic neuronal progenitors (Atoh1) (Fig 3D). The Cell Seek software identifies these cells as early glial cells (Fig 3D and S4B Fig).

PA-DR genes exhibited enrichment along specific cell lineages. Pax3, the most significantly overexpressed gene in *BRAF* fusion-positive infratentorial tumors, together with Irx5 and Irx2, exhibited expression across both early progenitors, GABA progenitors, and astrocytes (S5A to S5C Fig), with little expression in cells from earlier or later cell types, suggesting that Pax3 plays an important role at this key developmental stage. Similarly, Ascl1 and Pbx3 were expressed in earlier and later progenitors, respectively, while the remaining PA-DR genes were expressed in cells of the early astrocytic lineage (Meis1, Klf15, Msx2) (S5A to S5C Fig).

Computation of transcription factor networks (i.e. transcription factors whose expression are correlated) for these cell types recapitulated these two major lineages (GABA-ergic neuronal progenitors and glial cells). Seven out of eight PA-DR genes were computed to be in the same transcription factor network as these established markers of cell lineage (S6 Fig). Taken together, these data support a developmental paradigm operative within *BRAF* fusion-positive cerebellar pilocytic astrocytoma [42] which recapitulates this early developmental process whereby ventricular zone progenitors differentiate into GABA-ergic and glial lineages.

## Glial progenitors and astrocytes of the ventricular zone display evidence of MAPK activation

Given the central role of the canonical mitogen-activated protein kinase cascade in the pathogenesis of pilocytic astrocytoma [24], the expression of key markers of MAPK activity was interrogated within the developing cerebellum, namely, the transcription factors Fos and Jun.

## Overlap of Top Four Differentially Expressed PA-DR Genes

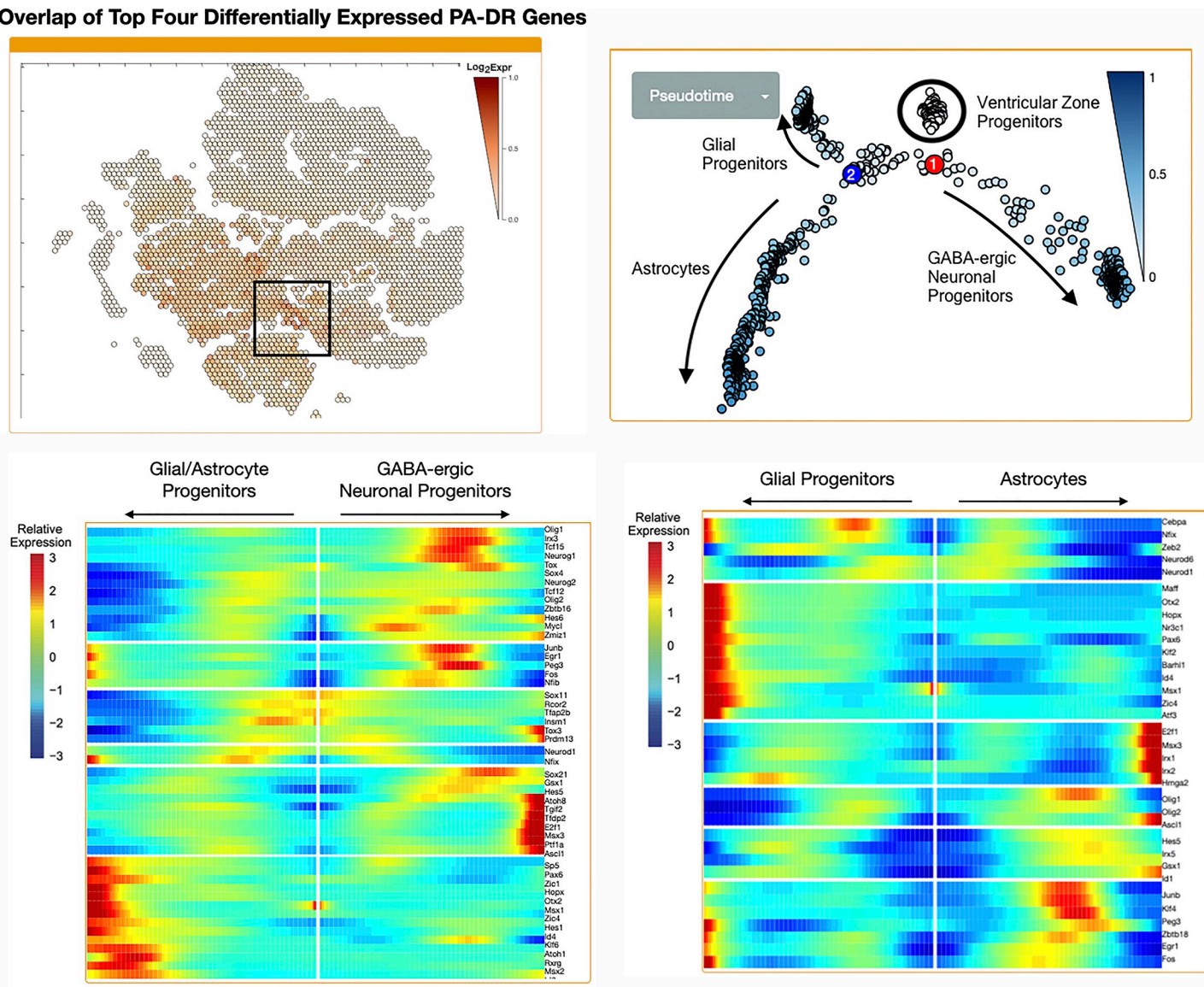

**Fig 3. PA-DR genes are co-expressed in early ventricular zone progenitors and these cells differentiate along GABA-ergic neuron, astrocyte, and glial progenitor lineages.** (A) Overlap of top four PA-DR genes in a single cell RNA-sequencing atlas of the developing mouse cerebellum. Each hexagon denotes a single cell or closely related group of cells plotted via t-SNE dimensionality reduction of the RNA sequencing data. Overlap was identified by stacking images from each of the four top differentially expressed PA-DR genes, revealing those cells with co-expression. The box denotes the region of strongest overlap. Expression scale is a heatmap depicting the $\log_2$ of the expression value. (B) Lineage analysis of early ventricular zone progenitor cells which express PA-DR genes reveals that these cells differentiate along three distinct lineages. Labels are added after manual curation of respective lineage gene expression. The blue color gradient refers to the position of each respective cell in "pseudotime,"–that is, degree of calculated differentiation (see reference for a description of the Monocle software and the concept of pseudotime [47]). The gradient of white to blue represents less and more differentiated cells, respectively. (C) Expression of genes which differentiate between the two main lineages marked by point 1 in (B), revealing one major lineage as GABA-ergic neurons. The middle of the figure represents the cell fate decision–each direction extending from the middle white line denotes one branch along the two different cell fate lineages. The blue-red gradient represents genes which are down- or up-regulated, respectively, as one travels along that lineage. (D) Expression of genes which differentiate between lineages marked by point 2 in (B). The trajectory toward glial progenitor cells is marked by expression of stem cell-like markers (e.g. Pax6) while the other lineage is enriched for astrocyte markers. Colors and interpretation as in (C).

Early astrocytic progenitor cells, but not ventricular zone progenitors or GABA-ergic progenitors, exhibited strong expression of both Fos and Jun (Fig 4A to 4B), suggesting MAPK activation within these cells.

## Fos Expression

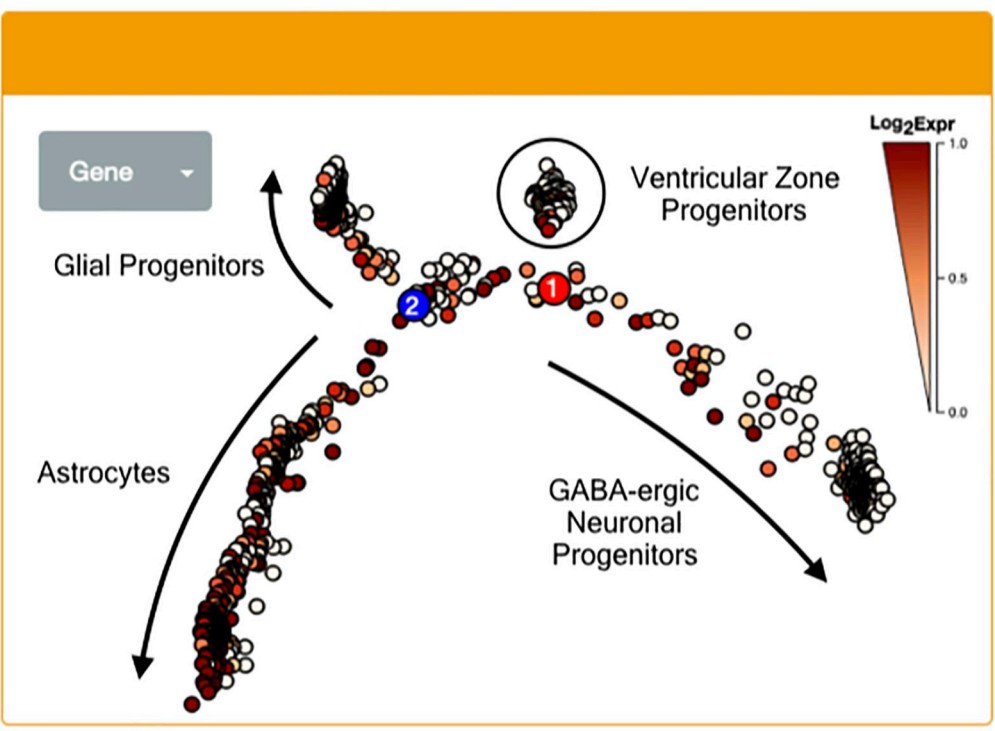

## Jun Expression

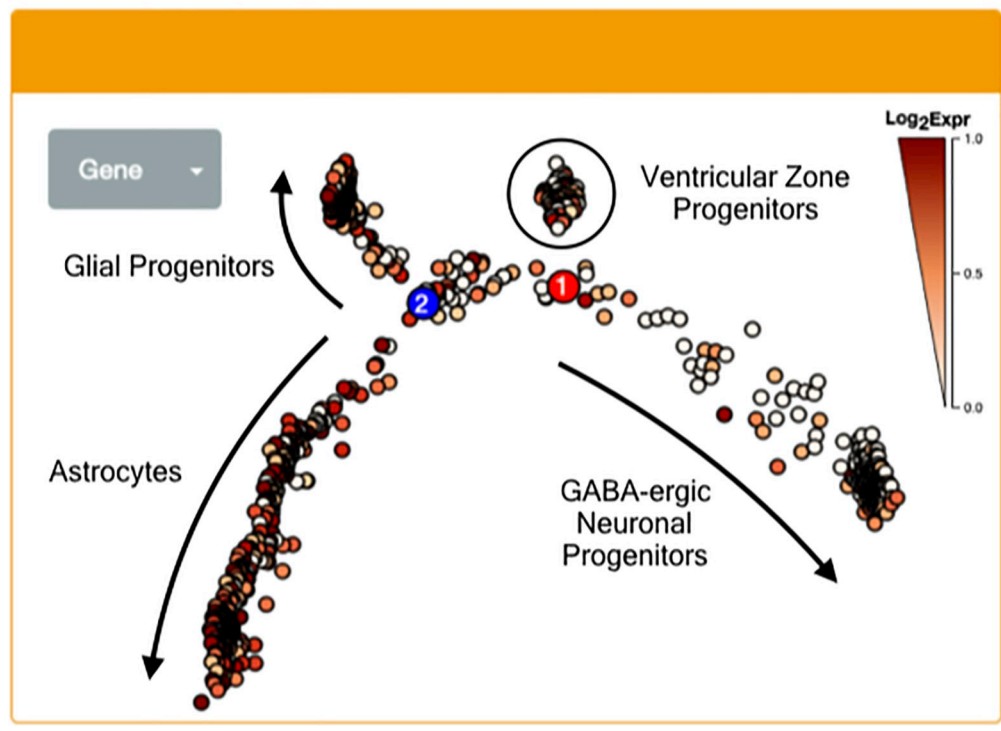

**Fig 4. Glial progenitors and differentiating astrocytes express markers of MAPK activation.** Expression of MAPK cascade transcriptional mediators Fos (A) and Jun (B) along Monocle lineages. Note the intense expression in intermediate progenitors (see also Fig 3D).

## Data from the developing human brain recapitulates expression of PA-DR genes within the developing cerebellum

Since all of the aforementioned developmental data is from mouse studies, validation of these findings in human data is needed. First the anatomic sites of PA-DR gene expression were identified within the developing human brain using BrainSpan (www.brainspan.org, Allen Institute). All PA-DR genes were expressed in the developing cerebellum, including many with exclusive expression therein (Fig 5A). Additionally, expression of PA-DR genes was sparse in postnatal brain samples, further underscoring the developmental nature of pilocytic astrocytoma. Second, in an organoid based model of brain development [40], PA-DR genes were expressed along the hindbrain neuron and astrocytic developmental lineages (Fig 5B).

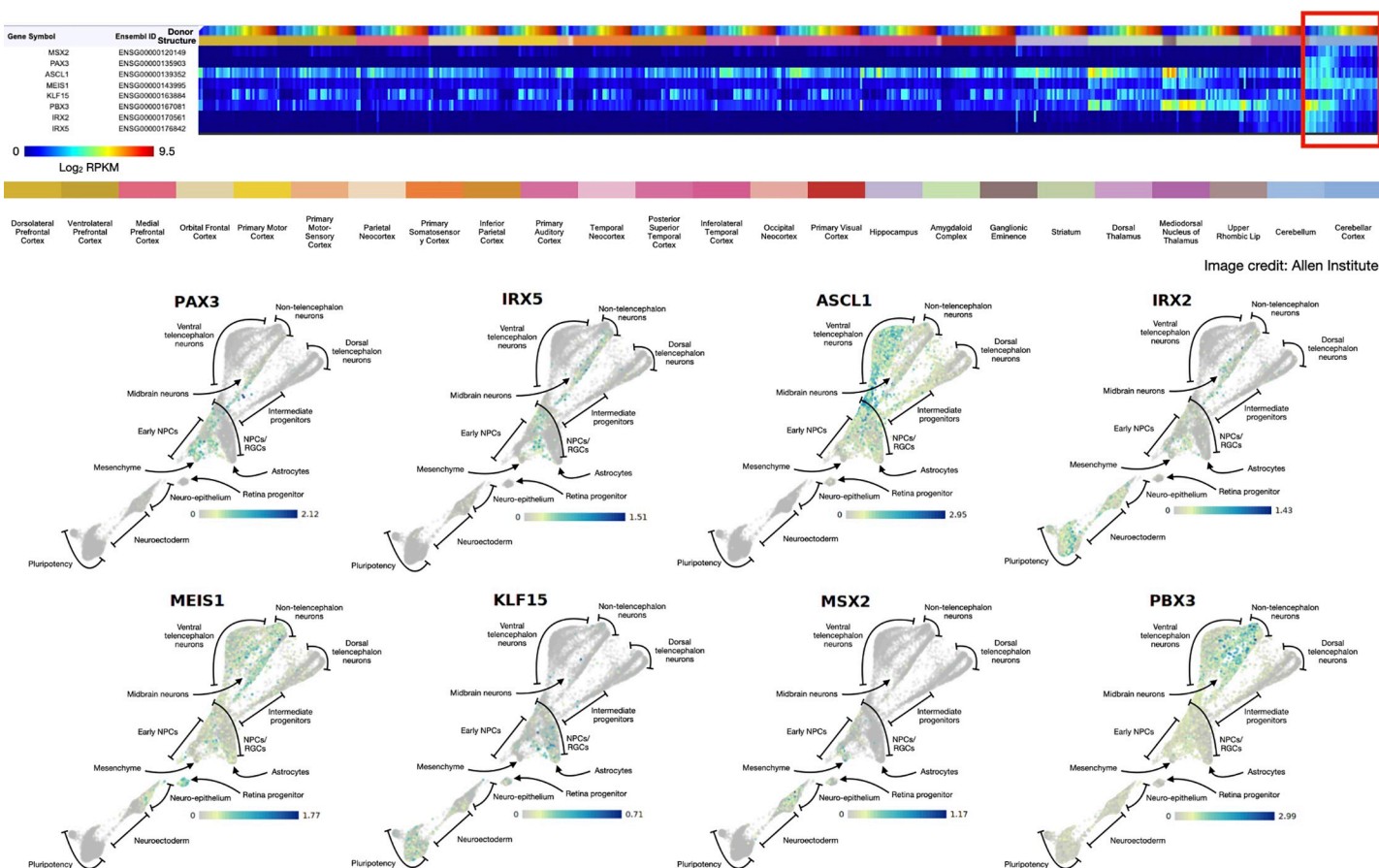

**Fig 5. Expression of PA-DR genes in the developing human brain recapitulates those patterns found in mice.** (A) Expression of PA-DR genes within the Developing Human Brain Atlas (Allen Institute). Image is ordered first by structure. Within each structure, each sample/donor is ordered by age (i.e. earliest samples are first within each structure subunit). The red box indicates cerebellar tissue. Expression values are in reads per kilobase per million (RPKM) as indicated in the heatmap. Note the exclusivity of PA-DR overlap within the early, developing cerebellum. (B) Expression of PA-DR genes within a pluripotent stem cell-derived developing brain organoid [40]. Data represent a t-SNE plot of single-cell RNA sequencing data from the developing brain organoid whereby each cell is a single point in the plot. Cell identities are indicated. For each PA-DR gene, the log2 expression value in each cell is shown by the color gradient. Note the trajectory of cells expressing PA-DR genes–most notably, Pax3 and Irx5, the most highly upregulated PA-DR genes in cerebellar pilocytic astrocytoma–are chiefly along the astrocyte and hindbrain neuronal lineages, recapitulating an identical developmental trajectory of PA-DR gene expressing cells in mice.

Taken together, these data support similar developmental trajectories and spatiotemporal localization of human cells expressing PA-DR genes as that found in the developing mice brain.

Thus, in conclusion, these data support early ventricular zone progenitor cells on the cusp of their GABA-ergic/astrocytic differentiation point as the cell of origin for *BRAF* fusion-positive cerebellar pilocytic astrocytoma.

## Discussion

Many pediatric cancers are characterized by perturbations and co-opting of normal developmental pathways that drive tumor development and growth. Understanding those pathways helps reveal the underlying tumor biology, shed light on predisposing factors, and suggests therapeutic approaches. The data presented herein map developmental genes overexpressed in *BRAF* fusion-positive cerebellar pilocytic astrocytoma to the normal developmental pathway of ventricular zone progenitor cells, suggesting that these cells represent the cell-of-origin for this tumor.

The unique temporal profiles of PA-DR genes within the Cell Seek atlas is of particular interest, given the documented developmental paradigm which has been described in pilocytic astrocytoma. Reitman and colleagues describe a developmental program operative within pilocytic astrocytoma tumors whereby neoplastic cells differentiate along three separate lineages [42]. The data presented here synergize therewith, mapping a nearly identical developmental paradigm occurring in normal cerebellar development (Fig 6). Thus, it is likely that the developmental program described in pilocytic astrocytoma tumors is a recapitulation of the normal pathway of cerebellar ventricular zone development described in this paper. Based upon this, it is tempting to speculate that *BRAF* fusion within a ventricular zone progenitor cells blocks normal differentiation "locking" the cell into a permanent embryonic state, similar to that process described in pediatric leukemia.

## Cerebellar Ventricular Zone Developmental Trajectory

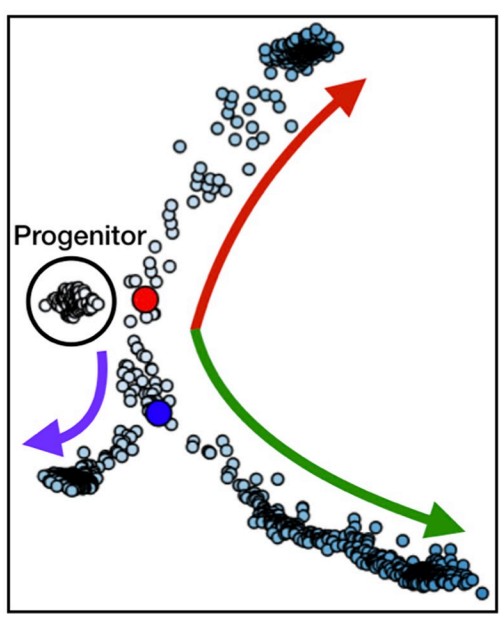

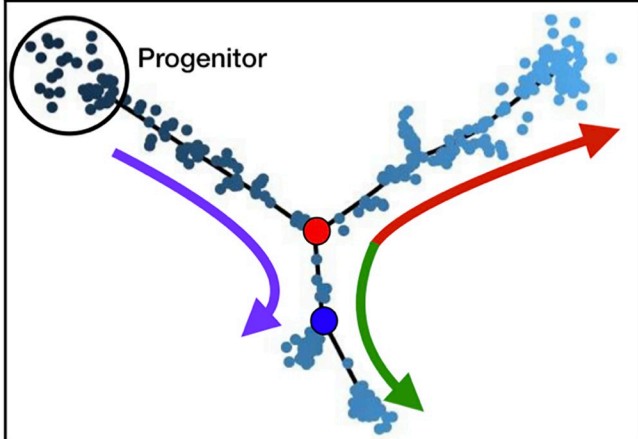

Pilocytic Astrocytoma Cellular Trajectory

Adapted from Reitman, et al. 2019

**Fig 6. Pilocytic astrocytoma cells recapitulate the developmental trajectory of ventricular zone progenitor cells.**

Data from Reitman, et al. 2019 [42] shows that the neoplastic cells of pilocytic astrocytoma differentiate along three distinct lineages. Early ventricular zone progenitor cells show very similar developmental architecture, differentiating along three distinct lineages, suggesting that pilocytic astrocytoma recapitulates this normal developmental pathway. The colored arrows are designed to draw attention to the very similar architecture of developmental trajectories between the cerebellar ventricular zone and pilocytic astrocytoma cells: namely, two cell fate decision points ultimately resulting in three different cell types.

Similar to the current study, data from Vladoiu and colleagues implicates the ventricular zone as the origin for cerebellar pilocytic astrocytoma [48]. The current study supports and extends those findings, further defining the developmental trajectory of the cerebellar ventricular zone, demonstrating MAPK pathway activation within ventricular zone-derived glial progenitor cells, and, uniquely, mapping these transcriptional and developmental data to the developing human brain.

Though de-differentiation of a mature cell type induced by *BRAF* fusion is not excluded, these data raise the intriguing possibility that *BRAF* fusion-positive cerebellar pilocytic astrocytoma has a prenatal origin, similar to other pediatric cancers [1]. For example, using newborn blood spots, several studies have shown that for children who develop acute lymphoblastic leukemia, the leukemic clone is present at birth [2–5]. Notably, however, not all neonates who have a leukemic clone at birth will go on to develop childhood leukemia, indicating that other genetic and environmental factors influence disease risk [6, 49]. It is tempting to speculate that a similar paradigm is true of *BRAF* fusion-positive cerebellar pilocytic astrocytoma. Given the lack of tissue for examining such a question and the obvious unethical nature of obtaining such samples, future studies examining this question will have to rely on other markers (e.g. circulating tumor cells).

This study has several limitations. First, it is possible that PA-DR genes simply represent gene expression of hindbrain neurons trapped within the tumor tissue rather than the tumor cells themselves. However, since most PA-DR genes are not expressed postnatally (see Fig 5A), this is unlikely.

Second, being *in silico*, while these data show consistency of PA-DR gene localization within the developing cerebellum across a variety of murine and human datasets, it cannot causally identify cerebellar ventricular zone progenitor cells as the cell-of-origin for cerebellar pilocytic astrocytoma. One approach to validating these findings *in vivo* would be to identify whether *BRAF* fusion alone within ventricular zone progenitor cells can lead to tumorigenesis. Given the relative impotence of *BRAF* fusion to cause neoplasia in other cell types [50], it would be fascinating to determine whether this genetic alteration is more oncogenic in the cellular context of ventricular zone progenitor cells.

Third, admittedly, the number of genes used in the analyses described herein is small; several factors contribute to this limitation. First, very few genes were differentially expressed between supra- and infra-tentorial tumors. Moreover, given the nature of the analyses, specific expression within the developing brain was required, further restricting the number of genes to analyze. Nevertheless, the consistency of spatio-temporal localization of even this small set of genes across multiple datasets partially mitigates this limitation. Moreover, several PA-DR genes, most notably Pax3 and the members of the Iroquois family (Irx5 and Irx2), have consistently been reported to be differentially methylated [25, 51, 52] and expressed at both the mRNA [25, 45, 51, 53] and protein levels [53] in infratentorial tumors, strengthening the selection of genes used in this study.

Fourth, this analysis is restricted to a single subtype of pilocytic astrocytoma; namely, those occurring in the cerebellum with a *BRAF* fusion. It is unlikely that this data can be extended to pilocytic astrocytomas with different drivers or those occurring in different locations, as those

tumor entities likely have a distinct cell-of-origin. However, given that the cerebellum is the most common location for pediatric pilocytic astrocytoma, and that *BRAF*-fusion is the most common driver, this data is highly relevant. In addition, the approach described in this paper, when coupled with appropriate developmental data, could be applied to pilocytic astrocytomas with different driver mutations, further defining the cellular and developmental origins of pilocytic astrocytoma.

In conclusion, with the above limitations in mind, these data provide compelling evidence for cerebellar ventricular zone progenitor cells as the cell-of-origin for *BRAF* fusion-positive cerebellar pilocytic astrocytoma. The current findings provide a key first step toward future validation research that may ultimately guide improved and targeted treatment for *BRAF* fusion-positive cerebellar pilocytic astrocytoma.

## Supporting information

**S1 Data.**
(PDF)

**S1 Table. List of genes differentially expressed between BRAF fusion-positive infratentorial vs. supratentorial tumors with accompanying statistical data.**
(XLSX)

**S2 Table. Table detailing the expression or lack thereof of PA-DR genes in three regions of the developing cerebellum.**
(XLSX)

**S1 Fig. PA-DR genes are expressed early in development of the mouse brain.** Expression for each PA-DR gene in the Allen Developing Mouse Brain Atlas. Note the particular enrichment on embryonic days 13.5 and 15.5 for all genes.
(TIFF)

**S2 Fig. Neurodevelopmental-related genes overexpressed in infratentorial pilocytic astrocytoma are expressed in rhombomere 1.** (A-D) Saggital and coronal images of the developing mouse brain with individual gene expression as marked. The purple highlight marks rhombomere 1. The white asterisk marks the cerebellar anlage. Many PA-DR genes exhibit a morphogenic gradient. Six out of eight PA-DR genes are expressed within rhombomere 1. Four are also expressed rostrally and five are also expressed caudally, suggesting rhombomere 1 as the region of overlap for these morphogenic gradients.
(TIFF)

**S3 Fig. PA-DR genes are enriched in early ventricular zone progenitors isolated from embryonic day 13–15.** (A) t-SNE plots of single cell gene expression data as in Fig 3A for each PA-DR gene. (B) Developmental day of isolation for single cells shown in Fig 3A. The box denotes the region of overlap of top four PA-DR genes, showing these cells are isolated from embryonic days 13–15. (C) Cell Seek derived cell type for single cells shown in Fig 3A. Based on expression of known cellular markers, cells co-expressing PA-DR genes are identified as early ventricular zone progenitor cells, GABA-ergic neurons, glia, and astrocytes.
(TIFF)

**S4 Fig. Lineage analysis of single cells from the developing mouse cerebellum which co-express PA-DR genes.** (A) Selection of cells used for subsequent lineage analysis. Bolded hexagons indicate cells which were selected while grayed out hexagons indicate cells which were excluded. (B) Cell seek derived cell types plotted along Monocle derived lineages revealing

three main cell types are derived from early ventricular zone progenitor cells: GABA-ergic neuronal progenitors, glial precursor cells, and astrocytes.
(TIFF)

**S5 Fig. PA-DR genes are individually enriched along certain lineages derived from ventricular zone progenitor cells.** (A) Zoomed region of interest from Fig 3A showing cell type for those cells with strongest overlap in expression of PA-DR genes. (B) Individual PA-DR gene expression for region of interest. Note the temporal relationship and lineage-specific expression of each PA-DR gene (C) Expression data for each PA-DR gene is shown along the Monocle-derived lineages. Note the enrichment of Pax3, Irx5, and Irx2 along all lineages. Ascl1 is enriched for early ventricular zone progenitor cells. Meis1, Klf15, and Msx2 are enriched along the glial progenitor and astrocytic lineages. Pbx3 is expressed chiefly in GABA-ergic neuron progenitor cells.
(TIFF)

**S6 Fig. Transcription factor correlation network reinforces cell developmental trajectories and places PA-DR genes within the same functional network as known regulators of cellular development.** Note that seven out of eight PA-DR genes are represented within the transcription factor network and localization therein recapitulates expression patterns/cell lineage restriction shown in S5 Fig.
(TIFF)

# Acknowledgments

Thanks to Kurt Showmaker, Ph.D. for careful review of the statistical and bioinformatic methodology used. The authors would like to thank the following persons for reviewing the manuscript and providing helpful feedback: Michael J. Ryan, Ph.D.; Cynthia Karlson, Ph.D.; and Jason Engel, Ph.D. Dr. Younes issues a special thanks to his wife, Hannah Younes, for her patience, kindness, helpful comments, and steadfast support.

# Author Contributions

**Conceptualization:** Subhi Talal Younes.

**Data curation:** Subhi Talal Younes.

**Formal analysis:** Subhi Talal Younes.

**Investigation:** Subhi Talal Younes.

**Methodology:** Subhi Talal Younes.

**Project administration:** Subhi Talal Younes.

**Resources:** Subhi Talal Younes.

**Software:** Subhi Talal Younes.

**Supervision:** Betty Herrington.

**Validation:** Subhi Talal Younes, Betty Herrington.

**Visualization:** Subhi Talal Younes.

**Writing – original draft:** Subhi Talal Younes.

**Writing – review & editing:** Subhi Talal Younes, Betty Herrington.

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
