## [Decision Letter · Decision Letter 0]

29 Jul 2020

PONE-D-20-17252

In Silico Analysis Identifies a Putative Cell-of-Origin for BRAF Fusion-Positive Cerebellar Pilocytic Astrocytoma

PLOS ONE

Dear Mr. Younes,

Thank you for submitting your manuscript to PLOS ONE. After careful consideration, we feel that it has merit but does not fully meet PLOS ONE’s publication criteria as it currently stands. Therefore, we invite you to submit a revised version of the manuscript that addresses the points raised during the review process.

After careful revision of your manuscript although very interesting several issues arise that should be address;

First please take into account the comments of the reviewer. SEcond the paper could use further review by a biostatistician, as the methodology used is complex.

I would also advice you to ask a further review by a senior colleague that could give further input. This is a recommendation not a must for the revision

We look forward to receiving your revised manuscript.

Kind regards,

Marta M. Alonso, PhD

Academic Editor

PLOS ONE

Journal Requirements:

Reviewers' comments:

Reviewer's Responses to Questions

**Comments to the Author**

1. Is the manuscript technically sound, and do the data support the conclusions?

Reviewer #1: Partly

2. Has the statistical analysis been performed appropriately and rigorously? 

Reviewer #1: I Don't Know

3. Have the authors made all data underlying the findings in their manuscript fully available?

Reviewer #1: Yes

4. Is the manuscript presented in an intelligible fashion and written in standard English?

Reviewer #1: Yes

5. Review Comments to the Author

Reviewer #1: In this study, Younes investigates the cell or origin for pediatric cerebellar juvenile pilocytic astrocytoma through an in silico approach. This is an ambitious study and carries relevance for the study of pediatric low-grade glioma. However, the study and especially the figures are often confusing and challenging to interpret. The lack of in vitro or in vivo validation of the findings is also a major issue. Additional issues are raised below.

Introduction:

- Line 73 would say radiation or chemotherapy, since both would rarely be necessary in these tumors.

- Would add a section on the known molecular alterations common to cerebellar pilocytic astrocytomas

Methods:

- Line 87, translocation -- does this refer to BRAF fusions?

- Need to define PA-DR at first mention in this section, not in the middle of Results

- There needs to be more justification of the use of a mouse atlas.

Results

- Line 162: Again, would change to BRAF fusion (throughout manuscript)

- The legend for Figure 2 is unclear -- what do the yellow dots represent?

- Figure 3 is also unclear -- color gradients need to be better defined, and images need to be labeled; screen captures from analysis programs generally are not good manuscript figure panels without editing to clarify and remove some unnecessary portions (e.g. a "Refresh" button)

- Wouldn't the lack of transcriptional upregulation of MAPK markers within the developmental cerebellum be because the fusion protein deregulating the pathway in tumors is not present?

- Again, Figure 5 is not clear and needs to be better explained in both the figure (via labels) and legend.

- These findings could likely be validated on a protein level through IHC/IF in human tumor samples, which would substantially strengthen the conclusions.

Discussion

- Figure 6: While usually figures should be associated with Results, this may be acceptable if allowed by editors; figure again needs to be better explained, however -- what do the colored arrows represent?

- More discussion is needed regarding the use of investigating markers from human tumor samples in a mouse brain atlas and then revalidating with a human atlas -- why not just use the human brain atlas throughout? This may be justifiable in terms of these developmental atlases not being available for humans, but this merits discussion.

6. PLOS authors have the option to publish the peer review history of their article (what does this mean?). If published, this will include your full peer review and any attached files.

Reviewer #1: No

---

## [Author Response · Author response to Decision Letter 0]

21 Oct 2020

Response to Reviewers

We would like to thank the reviewer and the editor for their thoughtful review and suggestions for improvement. In response, the manuscript has been extensively revised to incorporate the aforementioned reviewer and editor comments.

Editor Comments

Comment: The paper could use further review by a biostatistician, as the methodology used is complex.

Response: The methodology has been reviewed by Dr. Kurt Showmaker, a bioinformatician with expertise in differential gene expression analyses. His comments have been used to revise the description of the methodology used.

Comment: I would also advise you to ask a further review by a senior colleague that could give further input.

Response: Dr. Betty Herrington, a pediatric neuro-oncologist, has carefully reviewed the manuscript and analyses. Based on her contributions, she has been added as an author on the manuscript.

Reviewer Comments

Comment: In this study, Younes investigates the cell or origin for pediatric cerebellar juvenile pilocytic astrocytoma through an in silico approach. This is an ambitious study and carries relevance for the study of pediatric low-grade glioma. However, the study and especially the figures are often confusing and challenging to interpret. The lack of in vitro or in vivo validation of the findings is also a major issue.

Response: The figures and accompanying text have been extensively revised in order to make their interpretation clearer. In particular, the figure legends have been expanded to more carefully walk the reader through the data presented. Regarding the lack of in vitro or in vivo validation, the discussion has been updated to address these potential shortcomings. Most notably, potential avenues of in vivo validation have been proposed.

Comment: Line 37 should say radiation or chemotherapy, since both would rarely be necessary in these tumors.

Response: The line in question has been updated to correctly state that radiation or chemotherapy would be used for the treatment of pilocytic astrocytoma.

Comment: Would add a section on the known molecular alterations common to cerebellar pilocytic astrocytoma.

Response: A paragraph commenting on the known molecular alterations in cerebellar pilocytic astrocytoma has been added to the introduction.

Comment: Line 87, “translocation” – does this refer to BRAF fusions?

Response: This line did indeed seek to refer to BRAF fusion. The entire manuscript has been updated to properly state, “BRAF fusion.”

Comment: Need to define “PA-DR” at first mention in this section, not in the middle of Results.

Response: A line defining the designation of PA-DR has been added to the methods section

Comment: There needs to be more justification of the use of a mouse atlas.

Response: Several lines have been added explaining why we chose to utilize mouse atlases first. Briefly, as the reviewer suggests in a later comment, gene expression atlases of the developing human brain are sparse. On the other hand, mouse developmental atlases are more readily available with more robust data. Additionally, a section has been added to the discussion to more fully justify the use of a mouse gene expression atlas.

Comment: Line 162: Again, would change to BRAF fusion (throughout manuscript)

Response: All references throughout the manuscript have been changed to read, “BRAF fusion.”

Comment: The legend for Figure 2 is unclear – what do the yellow dots represent?

Response: The legend for Figure 2 has been revised to explain what is depicted.

Comment: Figure 3 is also unclear – color gradients need to be better defined, and images need to be labeled; screen captures from analysis programs generally are not good manuscript figure panels without editing to clarify and remove some unnecessary portions (e.g. a “refresh” button).

Response: Figure 3 has been extensively revised. The legend has been expanded and the figures themselves have been annotated to better direct the reader to the key data points.

Comment: Wouldn’t the lack of transcriptional upregulation of MAPK markers within the developmental cerebellum be because the fusion protein deregulating the pathway in tumors is not present?

Response: The results section in question has been revised to address a comment by the editor. Namely, the editor requested that the comment, “data not shown,” be removed from the manuscript. With the removal of that sentence, the reviewer’s comment has been relieved.

Comment: Again, figure 5 is not clear and needs to be better explained in both the figure (via labels) and legend.

Response: Figure 5 has been extensively revised and annotated to better convey the key data points.

Comment: These findings could likely be validated on a protein level through IHC/IF in human tumor samples, which would substantially strengthen the conclusions.

Response: A review of the literature was performed. Protein-level expression data could only be identified for a single PA-DR gene, namely, Pax3, confirming that it is overexpressed in cerebellar pilocytic astrocytoma. Unfortunately, a lack of funding precludes the ability to interrogate other markers.

Comment: Figure 6: While usually figures should be associated with Results, this may be acceptable if allowed by editors; figure again needs to be better explained, however -- what do the colored arrows represent?

Response: The figure legend has been revised.

Comment: More discussion is needed regarding the use of investigating markers from human tumor samples in a mouse brain atlas and then revalidating with a human atlas -- why not just use the human brain atlas throughout? This may be justifiable in terms of these developmental atlases not being available for humans, but this merits discussion.

Response: The discussion has been revised to more clearly state our rationale for having used mice atlases.

---

## [Decision Letter · Decision Letter 1]

3 Nov 2020

PONE-D-20-17252R1

In Silico Analysis Identifies a Putative Cell-of-Origin for BRAF Fusion-Positive Cerebellar Pilocytic Astrocytoma

PLOS ONE

Dear Dr. Younes,

Thank you for submitting your manuscript to PLOS ONE. After careful consideration, we feel that it has merit but does not fully meet PLOS ONE’s publication criteria as it currently stands. Therefore, we invite you to submit a revised version of the manuscript that addresses the points raised during the review process.

Please, address the following minor concerns: Improve the color keys to the color gradients in panels 3C, 3D, and 5A on the figures themselves to assist readers in interpreting these figures; 3C-D could likely also use text and/or arrows on the panels themselves to help with interpretation

We look forward to receiving your revised manuscript.

Kind regards,

Marta M. Alonso, PhD

Academic Editor

PLOS ONE

Reviewers' comments:

Reviewer's Responses to Questions

**Comments to the Author**

1. If the authors have adequately addressed your comments raised in a previous round of review and you feel that this manuscript is now acceptable for publication, you may indicate that here to bypass the “Comments to the Author” section, enter your conflict of interest statement in the “Confidential to Editor” section, and submit your "Accept" recommendation.

Reviewer #1: (No Response)

2. Is the manuscript technically sound, and do the data support the conclusions?

Reviewer #1: Yes

3. Has the statistical analysis been performed appropriately and rigorously? 

Reviewer #1: Yes

4. Have the authors made all data underlying the findings in their manuscript fully available?

Reviewer #1: Yes

5. Is the manuscript presented in an intelligible fashion and written in standard English?

Reviewer #1: Yes

6. Review Comments to the Author

Reviewer #1: The authors have done well in revising the manuscript and have addressed the majority of my concerns. While the figure captions are improved, I would still like to see better keys to the color gradients in panels 3C, 3D, and 5A on the figures themselves to assist readers in interpreting these figures; 3C-D could likely also use text and/or arrows on the panels themselves to help with interpretation.

7. PLOS authors have the option to publish the peer review history of their article (what does this mean?). If published, this will include your full peer review and any attached files.

Reviewer #1: No

---

## [Author Response · Author response to Decision Letter 1]

3 Nov 2020

Response to Reviewers

We would like to thank the reviewer for their thoughtful review and suggestions for improvement. In response, the manuscript has been revised to incorporate the aforementioned reviewer comments.

Reviewer Comments

Comment: The authors have done well in revising the manuscript and have addressed the majority of my concerns. While the figure captions are improved, I would still like to see better keys to the color gradients in panels 3C, 3D, and 5A on the figures themselves to assist readers in interpreting these figures; 3C-D could likely also use text and/or arrows on the panels themselves to help with interpretation.

Response: Figures 3C, 3D, and 5A have each been updated to include a color gradient key. In addition, arrows and text descriptions have been added to figures 3C and 3D to aid the reader in interpreting which direction represents which cell lineage.

---

## [Editor Report · Decision Letter 2]

4 Nov 2020

In Silico Analysis Identifies a Putative Cell-of-Origin for BRAF Fusion-Positive Cerebellar Pilocytic Astrocytoma

PONE-D-20-17252R2

Dear Dr Younes,

We’re pleased to inform you that your manuscript has been judged scientifically suitable for publication and will be formally accepted for publication once it meets all outstanding technical requirements.

Kind regards,

Marta M. Alonso, PhD

Academic Editor

PLOS ONE
---

## [Editor Report · Acceptance letter]

6 Nov 2020

PONE-D-20-17252R2 

**In *Silico* Analysis Identifies a Putative Cell-of-Origin for *BRAF* Fusion-Positive Cerebellar Pilocytic Astrocytoma**

Dear Dr. Younes:

I'm pleased to inform you that your manuscript has been deemed suitable for publication in PLOS ONE. Congratulations! Your manuscript is now with our production department. 

Kind regards, 

on behalf of

Dr. Marta M. Alonso 

Academic Editor

PLOS ONE